# FFAEVAL: Evaluating Dialogue System via Free-For-All Ranking

**Zeyao Ma**[1,2,*], **Zijun Yao**[3,*], **Jing Zhang**[1,2,†], **Jifan Yu**[3],
**Xiaohan Zhang**[4], **Juanzi Li**[3], **Jie Tang**[3]

[1]School of Information, Renmin University of China, Beijing, China
[2]Engineering Research Center of Database and Business Intelligence, MOE, China
[3]Department of Computer Science and Technology, Tsinghua University, Beijing, China
[4]Zhipu.AI
{zeyaoma, zhang-jing}@ruc.edu.cn, {yaozj23, yujf21}@mails.tsinghua.edu.cn
{xiaohan.zhang}@aminer.cn, {juanzi, jietang}@tsinghua.edu.cn

## Abstract

Evaluating open-domain dialogue systems is currently an open question. Automatic evaluation metrics have shown poor correlation with human assessment in dialogue generation tasks. Human evaluation, which involves annotators for multi-dimension scoring, is trustworthy but time-consuming. In this work, we propose FFAEVAL, a reliable and efficient human **EVAL**uation framework using **F**ree-**F**or-**A**ll ranking approach. By sharing the dialogue history, the framework enables annotators to converse with multiple dialogue systems simultaneously in a single-blind, multi-turn manner. The subsequent free-for-all allows annotators to select the most favourable model in each turn from among all the participating dialogue systems. The final performance of each model is represented by calculating the TrueSkill score derived from the free-for-all competition. Our empirical study on English and Chinese dialogue systems demonstrates that FFAEVAL achieves a strong correlation with score-based human assessment compared to existing evaluation methods. We further prove the efficiency and stability of our framework in additional experiments. The source code and data are available on Github[‡].

## 1 Introduction

Designing reliable and efficient evaluation methods serves as a beacon for the improvement of open-domain dialogue systems. It becomes increasingly challenging in this era dominated by dialogue systems that are typically built upon generative large language models (LLMs) (Shuster et al., 2022; Zhang et al., 2023). These systems generate open-ended responses from various perspectives, each

---

[*]Equal Contribution.
[†]Corresponding author.
[‡]https://github.com/RUCKBReasoning/FFAEval

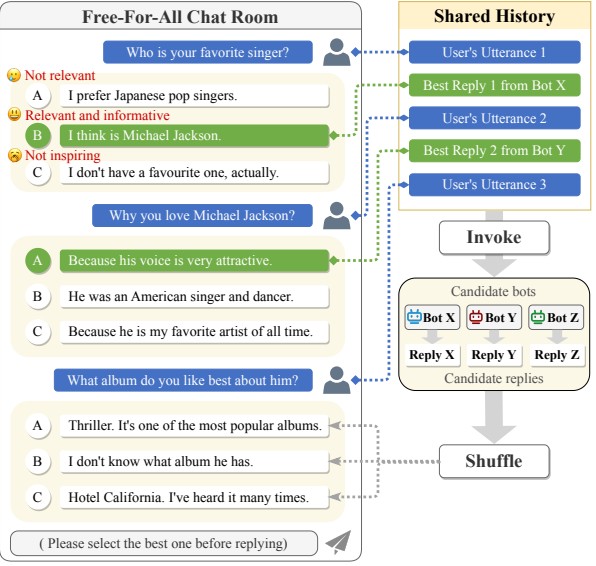

Figure 1: The evaluation process of a single Free-For-All match.

carrying unique information and purposes. How to accurately and impartially evaluate these diverse responses remains a pressing issue.

Existing methods for evaluating dialogue systems primarily fall into two categories: automatic evaluation and human evaluation. For **automatic evaluation**, metrics are derived by comparing the system-generated response with a pre-annotated utterance, *e.g.,* ROUGE (Lin, 2004), BERTScore (Zhang et al., 2020a), and BARTScore (Yuan et al., 2021). However, as there is no definitive 'correct' answer for a dialogue response, these metrics fail to recognize viable responses that diverge from the reference. While using LLMs to score responses can mitigate this issue (Bai et al., 2023; Chiang et al., 2023), they tend to bias towards self-similar responses (Liu et al., 2023b). Thus, human assessment, involving annotators interacting with the dialogue system and

rating its responses according to guidelines (Finch and Choi, 2020), remains the *de facto* gold standard.

Nevertheless, **human evaluation** presents its own challenges. It proves exceedingly difficult and time-consuming for annotators to complete questionnaire demanding subjective scores (Louviere et al., 2000). Recent approaches aim to approximate these scores by having annotators choose preferred models from a pair of options (Deriu et al., 2020; Zheng et al., 2023). However, the pairwise model comparison still requires a considerable volume of annotations to converge to a stable score.

In this paper, we present FFAEVAL, a human **EVAL**uation framework for dialogue system that uses **F**ree-**F**or-**A**ll ranking approach. The core idea is to enable annotators to converse with multiple dialogue systems simultaneously in a single-blind, multi-turn fashion by sharing the dialogue history. The subsequent free-for-all allows annotators to select the most favourable model in each turn from among all the participating dialogue systems. The final performance of each model is represented by calculating the TrueSkill (Herbrich et al., 2006) score derived from the free-for-all competition. We demonstrate FFAEVAL in Figure 1. Contrary to score-based human evaluation, FFAEVAL only requires selection, which is more efficient and less labour-intensive (Louviere et al., 2000).

FFAEVAL shows unique advantages to other selection-based evaluation methods for dialogue systems (*e.g.,* Chatbot Arena (Zheng et al., 2023)) from 3 aspects. First and most importantly, FFAEVAL maintains a shared dialogue history for every participating dialogue system, which effectively prunes the unfairness introduced when the annotator implicitly replies to one of the dialogue systems; Second, as opposed to pairwise comparison and conversation-wise selection, FFAEVAL compares all the candidate participates together for each turn of response, which makes the evaluation result of FFAEVAL converge faster and more fairly; Third, FFAEVAL derives the TrueSkill score from free-for-all ranking, which is less sensitive to the order of competition, making the ranking more robust.

We conduct experiments on both Chinese and English dialogue systems powered by LLMs. Compared to a series of automatic evaluations and other selection-based human evaluation methods, the evaluation score from FFAEVAL demonstrates its superiority in terms of consistency with score-based human evaluation results. We also conduct comprehensive analysis on FFAEVAL, and find that to obtain a trustful evaluation score upon 5 different English dialogue systems using FFAEVAL only requires 175 round of conversation. Moreover, FFAEVAL is less sensitive to match order. We further conduct an experiment to estimate the confidence intervals of both Chatbot Arena and our framework and discover that FFAEVAL is much more stable than Chatbot Arena.

## 2  FFAEVAL Framework

In general, FFAEVAL treats each multi-turn conversation as a multiplayer competition, in which participating dialogue systems seek to gain recognition from human users. FFAEVAL comprises three main components: (1) A **human-bot interaction** mechanism between evaluators and dialogue systems to obtain the competition ranking for each conversation; (2) A **dialogue history sharing** mechanism to ensure a fair context for response generation; (3) A **global ranking scoring** algorithm to produce the global ranking by synthesizing all the historical competition rankings. This rest of this section will introduce these modules in detail. The overall framework of FFAEVAL is demonstrated in Figure 2.

### 2.1  Human-Bot Interaction

Human users engage in three types of interaction with the dialogue system. First, they converse with all the dialogue systems simultaneously with multiple turns. Second, they are guided to choose the best model response for each conversational turn. Finally, they decide when to end the conversation, enabling FFAEVAL to finalize the ranking of each dialogue system based on that conversation.

**Multi-turn Human-Bot Conversation.** FFAEVAL prompts human users to initiate the dialogue. In each turn, the message of the user is appended to the end of the shared history. We structure the dialogue history into the following prompt template, which is then provided to each dialogue system as input to elicit their responses:

```
[User    Utterance 1] <SEP>
[Selected Response 1] <SEP>
[User    Utterance 2] <SEP>
[Selected Response 2] <SEP>
...
[User    Utterance n]
```

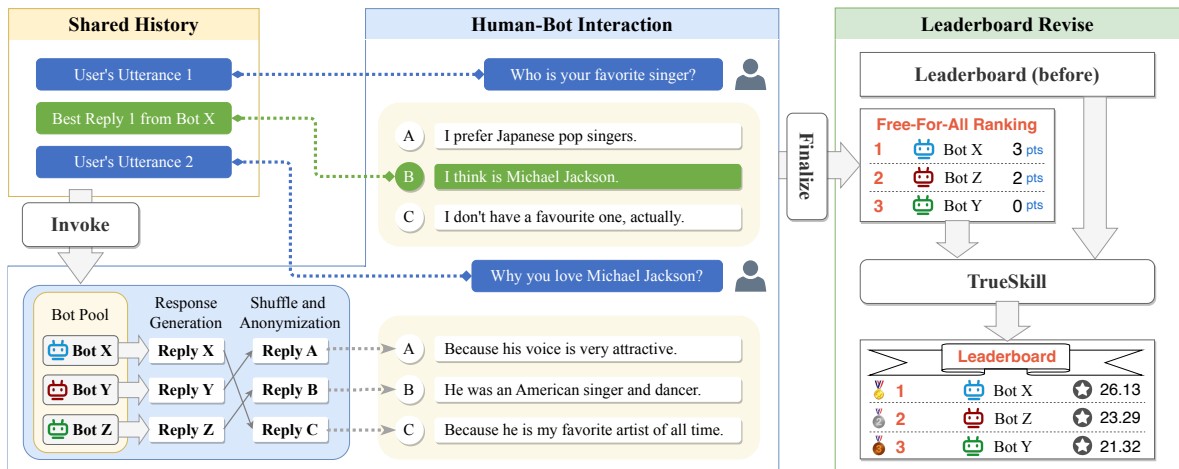

Figure 2: The general human evaluation framework for multiple dialogue systems using FFAEVAL.

The token <SEP> is replaced with model-specific tokens before being sent to the conversation model, *e.g.,* \n for BlenderBot and <sep> for EVA2.0. Meanwhile, template tokens within [Brackets] are replaced with corresponding content from the dialogue history. To speed up the response generation process, we employ multiple threads to gather responses from each model simultaneously.

**Best Response Selection.** After all the participated dialogue systems generate their utterance, the user is asked to choose the response that is most appropriate to continue the conversation. We implement a series of mechanisms to ensure fair evaluation. For anonymization, the responses are shuffled and post-processed before being shown to the user. During post-processing, FFAEVAL applies rules to identify and replace spans that could potentially reveal the identity of the dialogue system with placeholders. Moreover, to mitigate the primacy effect (Malhotra, 2008)—that is, users paying more attention to the first presented response—FFAEVAL presents all responses simultaneously.

Although *selecting appropriate response* is the easiest way to use FFAEVAL, it lacks a strict definition with explicit evaluation guideline. Thus, we also refer to the score-based human evaluation principles, and ask the annotators to consider the contextual coherence, informativeness of the response, and veracity of the content. Furthermore, multi-dimensional scoring puts constraints on annotators within pre-designed rules. In this way, FFAEVAL could not only approximate gold standard score-based evaluation methods, but also provides a better way to assess the overall ability of the dialogue model.

**Conversation Finalization.** In each dialogue turn, the dialogue system chosen by the user is awarded 1 point, while the others receive none. The user is free to end the conversation at any point. FFAEVAL accumulates these points to generate a leaderboard that ranks each dialogue system based on its score in the conversation. This leaderboard is then used to compute the global ranking score. It is worth noting that, we empirically find allowing users to select multiple responses has a slight negative impact on the model evaluation results because it becomes easier for many dialogues systems to receive similar points. Thus, FFAEVAL only prompts users to select exactly one model.

## 2.2 Shared Dialogue History

The shared dialogue history is the key component that enables user to converse with multiple dialogue system simultaneously. As different dialogue system generate replies with potentially different topics to the same human utterance, the dialogue system can encounter what we term *cascading bias*—dialogue systems that are not selected in the initial turn would have a contradictory dialogue history, which consequently cause them to perform poorly in future turns. For instance, as illustrated in Figure 1, without a shared dialogue history, two bots not selected in the first turn would be confused by the user response, which could lead to disorganized replies and, thereby, degrade the points earned by these two bots.

Technically, the shared dialogue history functions as a stack, and its push-back operation is triggered by two events. First, when the user finishes their input, their response is appended to the stack. Second, once the user selects a response from the presented choices, the response from the chosen

dialogue system is appended to the stack.

## 2.3 Global Ranking Score

The global ranking score quantifies the skill level of each model after a series of conversations.

**TrueSkill Estimation.** Inspired by algorithms tracking the skill level of players in games, FFAE-VAL calculates TrueSkill (Herbrich et al., 2006) score as the ranking score. Compared to other rating algorithms such as Elo (Glickman and Jones, 1999), which is used by Chatbot Arena, TrueSkill presents three major advantages: (1) TrueSkill is designed for free-for-all involving any number of players, whereas Elo is intended for 1v1 matches; (2) TrueSkill tracks the estimated variance of the score, providing a confidence interval for the estimated ranking score; (3) TrueSkill adjusts significantly when the algorithm is less certain, thus converging with fewer conversations.

In particular, TrueSkill models the ranking score of each participants as a Gaussian distribution with two parameters, namely, the mean score $\mu$ and the standard deviation $\sigma$. FFAEVAL initiate the parameters from $\mu = 25$, $\sigma = \frac{25}{3}$, as suggested by Herbrich et al. (2006). After each conversation generating its corresponding ranking result, TrueSkill uses Bayes' law to update the ranking score.

**Leaderboard.** After sufficient number of conversations, the score of each dialogue system is computed using the following equation:

$$S = \mu - k * \sigma \tag{1}$$

This is a conservative skill estimate that gives a lower bound for the possible real skill. FFAEVAL follows Herbrich et al. (2006) to sets $k = 3$. The leaderboard is updated by the order of skill score.

## 3 Experiments

We conduct extensive experiments to verify the effectiveness of FFAEVAL in evaluating generative dialogue systems.

### 3.1 Dialogue Systems to Evaluate

We deploy English dialogue systems on NVIDIA GeForce RTX 3090, utilizing their corresponding public source codes. These models include: *Seq2Seq* represents our implementation of the BART model (Lewis et al., 2020), which is fine-tuned on DailyDialog (Li et al., 2017). *DialoGPT* (Zhang et al., 2020b) fine-tunes the GPT-2 model (Radford et al., 2019) on Reddit comments

and contains 345M parameters. *BlenderBot-3B (BB-3B)* (Shuster et al., 2022), initially pre-trained on Reddit comments, is later fine-tuned on manually annotated dialogue data to improve performance across various domains, and possesses 3B parameters. We have also evaluated *BlenderBot-90M (BB-90M)*, a variant model with 90M parameters. *PLATO-XL* (Bao et al., 2022), trained on an extensive collection of social media data from the Internet, incorporates 11B parameters.

To affirm that our FFAEVAL remains effective across multiple languages, we have also conducted experiments on Chinese dialogue systems, which include: *CDial-GPT* (Wang et al., 2020), a GPT-structured model with 95.5M parameters, which is fine-tuned on the LCCC dataset. *EVA2.0* (Zhou et al., 2021), a generative conversation model with 2.8B parameters, trained using 1.4T dialogues sourced from social media. *PLATO-2* (Bao et al., 2021), a 336M language model, is trained using curriculum learning techniques on 1.2B examples from social media conversations. *XDAI* (Yu et al., 2022), a tuning-free dialogue model based on GLM-10B-Chinese, which additionally maintains an external knowledge base to enhance its capabilities. *GLM-Dialog* (Zhang et al., 2023), a retrieval augmented dialogue system with 10B parameters.

### 3.2 Baselines Evaluation Methods

We compare FFAEVAL against both automatic evaluation metrics and human evaluation methods.

For automatic metrics, we consider: **F1** and **BERTScore (BS)** (Zhang et al., 2020a) are reference-based metrics which compute the similarity between model-generated and human responses. **ChatMatch (CM)** (Yang et al., 2022) calculates multiple scores for conversation contents between two different dialogue systems. This pair-wise match generates an overall ranking via round-robin tournament.

For human evaluation, we mainly focus on selection-based methods. **Chatbot Arena (CA)** (Zheng et al., 2023) enables users to converse with two anonymous models side-by-side and vote for which one is better. The rating of each model is updated via the Elo rating system. **Preference ranking (PR)** is our implemented baseline, which is based on the method designed to collect comparison data to train reward model (Ouyang et al., 2022). It asks annotators to rank the generated responses conditioned on a same dialogue context

without multi-turn human-bot interactions. We utilize TrueSkill to calculate the ranking result of each conversation as the score of each dialogue system. PR is different from FFAEVAL in that it is static without shared dialogue history and can only evaluate single-round conversation.

## 3.3 Evaluation Details

We consider the following miscellaneous to setup experiments for FFAEVAL.

**Human Resources.** We recruit 20 undergraduate students, who use part time to annotate deployed dialogue systems with FFAEVAL. To ensure consistency among different human evaluation methods and the overall metric of the GS score, these annotators are asked to evaluate following the same unified guidelines as provided in Appendix A. Besides, all the human evaluation methods ask the annotators to evaluate the same number of turns of responses, which guarantees the convergence of all the evaluation methods.

**Data Resources.** The reference-based metrics are calculated on DailyDialog (Li et al., 2017) and Diamante (Lu et al., 2022) datasets for English and Chinese models, respectively. Both datasets are also used as seed topics to prompt annotators to start conversations.

**Gold Standard (GS) Score.** We consider human assessed scores to be the gold standard for evaluating dialogue systems, as this method is frequently employed by researchers (*e.g.*, (Bao et al., 2022), (Zhang et al., 2023)). In particular, on each occasion, we begin by randomly selecting an initial utterance from DailyDialog (Li et al., 2017), Dusinc (Zhou et al., 2022), and Diamante (Lu et al., 2022). After making necessary modifications, this utterance is subsequently employed by all dialogue systems to initiate human-bot conversation. Then, we ask another group of annotators to assign scores, including the overall performance and other dimensions. We use the overall scores as the gold standard and consider other dimensions as references in Appendix B. Each dialogue score is an averaged over three annotators.

We use Pearson's correlation to calculate interannotator agreement (IAA) score of ground truth annotation. The annotations are reliable, as the IAA scores for the Chinese and English experiments are 0.63 and 0.59, respectively. We calculate the correlation coefficients between each evaluation method and the ground truth, including Kendall

| Method | English | | Chinese | |
|---|---|---|---|---|
| | $\tau$ | $\rho$ | $\tau$ | $\rho$ |
| F1 | 40% | 27.3% | 20% | 17.5% |
| BERTScore | 0% | 5.2% | 40% | 30.2% |
| ChatMatch | $-20\%$ | $-38.4\%$ | 40% | 13.5% |
| Preference Ranking | 60% | 84.2% | 60% | 77.1% |
| Chatbot Arena | 60% | 55.0% | 80% | 83.5% |
| FFAEVAL | **100%** | **97.7%** | **100%** | **98.6%** |

Table 1: Overall correlation between the result of different methods and gold standard scores on English and Chinese experiment. $\tau$ and $\rho$ stand for Kendall ranking correlation and Pearson score correlation, respectively.

ranking correlation $\tau$ and Pearson correlation $\rho$. Methods with a higher correlation are considered to have better reliability.

## 3.4 Main Results

We are mainly concerned about two questions: Does FFAEVAL generate accurate rankings on the leaderboard, and to what extent does the ranking score of FFAEVAL correlates with the gold standard evaluation score?

**Leaderboard Rankings.** Figure 3 illustrates the ranking results of all dialogue systems under various evaluation methods, with the gold standard ranking featured in the far-right column. Our experimental findings suggest that FFAEVAL yields rankings identical to those of the score-based evaluations when assessing both English and Chinese models. As a comparison, both reference-based metrics—namely F1 and BARTScore—and reference-free metrics, *i.e.*, ChatMatch, do not succeed in accurately approximating the ranking from the gold standard rankings.

**Ranking Scores.** We further explore whether the TrueSkill score of FFAEVAL provides a detailed approximation from score-based human assessments. The correlation between various approximation scores and the gold standard score is presented in Table 1. We find that all methods based on human evaluation outperform automatic evaluation methods, emphasizing the importance of human involvement when assessing dialogue systems. Among all the human evaluation methods, FFAEVAL achieves the highest Pearson correlation at 97.7% in evaluating English models and 98.6% in evaluating Chinese models, which is a large margin surpassing all baseline methods.

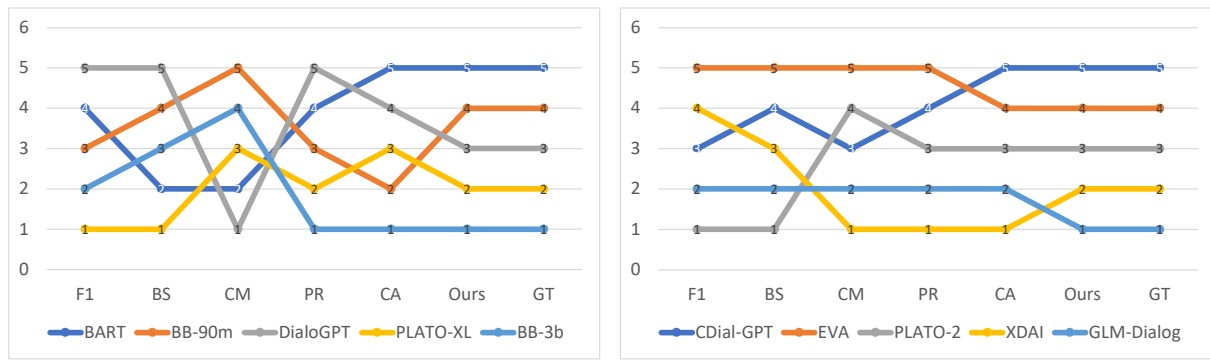

Figure 3: The ranking of dialogue systems provided by each methods. Left and right figure show the result of English and Chinese Experiment, respectively.

Of all evaluation methods, ChatMatch performs the worst, even demonstrating a negative correlation with the gold standard ranking results. This is due to the fact that ChatMatch evaluates dialogues between two dialogue systems, which often devolve into content lacking in information. This indicate that it is necessary to evaluate conversations between humans and dialogue systems, as adopted by FFAEVAL.

Preference Ranking, which ranks dialogue systems based on single-turn conversations, serves as an ablation experiment to evaluate the effect of multi-turn conversation in the assessment process. FFAEVAL achieves 13.5% and 21.5% higher Pearson correlation in English and Chinese, respectively, thereby underlining the importance of permitting a series of interactions between the annotators and the dialogue system.

We are also concerned about the comparison between FFAEVAL and Chatbot Arena, both of which transform preference-based human evaluation into competitions for dialogue systems. FFAEVAL outperforms Chatbot Arena by a margin of 42.7% in English and 15.1% in Chinese. The advantage of FFAEVAL lies in its free-for-all competition format, which exhibits no sensitivity to the competition order. This format allows for a more accurate approximation of the gold standard ranking score due to its smaller deviation.

## 4 Analysis

We analyze the characteristics of FFAEVAL and the feasibility of each designed components.

### 4.1 Efficiency Analysis

It is also important for FFAEVAL to provide highly efficient human evaluation. We measure efficiency from two aspects—the time required to evaluate an

| Method | Annotation Time | |
| --- | --- | --- |
| | English | Chinese |
| Preference Ranking | ~415 min | ~335 min |
| Chatbot Arena | ~665 min | ~500 min |
| Gold Standard | ~530 min | ~500 min |
| FFAEVAL | **~250 min** | **~300 min** |

Table 2: The annotation time of different human evaluation methods on English and Chinese experiments.

equivalent number of responses (Time Efficiency), and the number of responses needed for the scores to converge (Convergence Efficiency).

**Time Efficiency.** We conduct a fair time efficiency comparison among FFAEVAL and baselines by recording the time elapsed to annotate 2000 responses, which we empirically find it sufficient for all the methods converage to a stable score. For each human evaluation method, we log the entry and exit times of each annotator in the online system. We are able to calculate the time spent by each annotator and obtain the total annotation time through summation as shown in Table 2.

FFAEVAL demonstrated a significant improvement in time efficiency among all the evaluation methods. This is due to FFAEVAL allows the annotators to evaluate multiple dialogue systems simultaneously. Chatbot Arena uses much more time to evaluate the same amount of dialogues compared with our framework. This is due to the fact that Chatbot Arena only allows annotators to converse with two dialogue systems side-by-side. Moreover, in Chatbot Arena, the annotator must concentrate on two different dialogue contexts, whereas in FFAEVAL, only one shared history is required.

**Convergence Efficiency.** We conduct experiments to compare the convergence efficiency be-

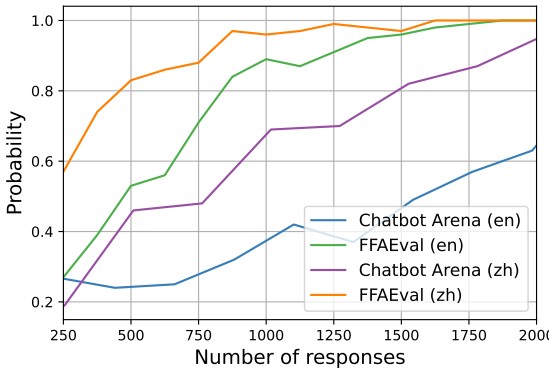

Figure 4: Ranking stability towards the number of responses. The x-axis represents the number of responses generated by dialogue systems. The y-axis represents the rate at which the same ranking is achieved across 100 repetitions.

tween free-for-all competitions, as adopted by FFAEVAL, and pair-wise competitions, as adopted by Chatbot Arena. In particular, we use FFAE-VAL and Chatbot Arena to evaluate dialogue systems until reaching a stable ranking result. From these annotation results, we randomly sample $n$ responses and use their ranking algorithm to produce the ranking result. The trait is repeated for 100 times to calculate the possibility that ranking with $n$ responses yields identical ranking as the stable result. We range $n$ from 250 to $2,000$, and plot the possibilities on Figure 4.

The convergence of FFAEVAL is faster than that of Chatbot Arena in both English and Chinese experiments. There are two possible factors that count for this result: (1) Compared to Chatbot Arena, which evaluates dialogue systems side-by-side, FFAEval evaluates all models in a single match, which is much more efficient and provides a more thorough comparison of dialogue systems. (2) The evaluation process of FFAEval is more fine-grained, as annotation is involved in each turn of conversation, whereas Chatbot Arena evaluates after several turns.

### 4.2 Sensitivity Analysis

In this section, we discuss the sensitivity of Chatbot Arena and FFAEVAL to match order. Specifically, we shuffle the competition results for $1,000$ times to estimate the confidence intervals. For fair comparison, the ranking scores are normalized to $[0, 1]$.

As shown in Figure 5, our framework is much more stable than Chatbot Area, as the values of FFAEVAL converge to a small range while the val-

ues of Chatbot Arena are relatively dispersed. This result demonstrates that FFAEVAL is insensitive to match order and thus more stable in comparison to Chatbot Arena, due to the free-for-all ranking of our framework, as the ratings of all dialogue systems are updated after each match.

### 4.3 Turn Number Analysis

In this section, we discover the influence of different maximum turn number on the final ranking result. Following the same sampling process in convergence efficiency analysis, we calculate the convergence result of free-for-all by setting different maximum number of turns.

Figure 6 presents that, free-for-all with only 1 and 2 turn is not able to achieve the same ranking with the original result. This is due to the fact that some dialogue systems excel at single-round conversations but perform poorly during long conversations. The convergence speed of conducting 5 turns in a free-for-all match is faster than 4 turns. As the maximum turns in a free-for-all match increase, the results of a single match can more accurately reflect the true level of each dialogue system. On the one hand, more turns can demonstrate the ability of the dialogue system to engage in multi-turn conversation. On the other hand, more turns can reduce the impact of possible annotation bias.

### 4.4 Responses Selection Mechanism Analysis

In FFAEVAL, the annotator only selects one response in each round of conversation, which may appear imprecise given that multiple optimal responses may exist. We modify the best response selection procedure to permit annotators to choose multiple responses for each dialogue turn. To maintain the shared dialogue history, when the annotator selects multiple responses, we select one at random and use it as the shared history.

In English experiment, this modified version of FFAEVAL yields the same ranking as the original, with a slight decrease in Pearson's correlation from $97.7\%$ to $94.6\%$. This result shows that only select one response is enough to achieve a satisfactory results. The decreased is caused by the fact that annotators tend to give credit to many dialogue systems, making the ranking score less discrimitive.

### 4.5 Shared Dialogue History Analysis

We finally conduct experiments to examine the effectiveness of the shared dialogue history. First,

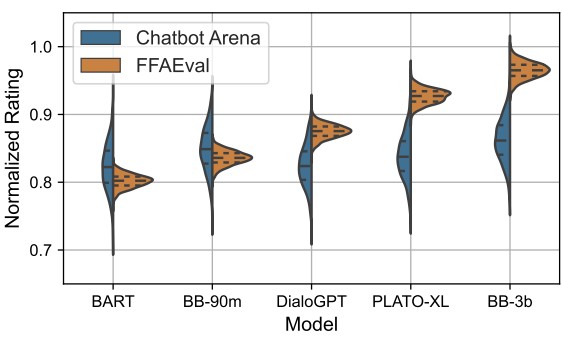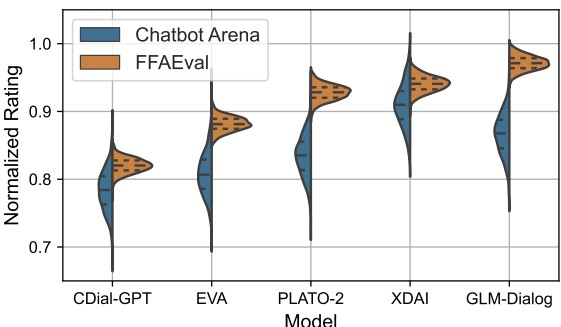

Figure 5: The confidence interval of the normalized ranking score between Chatbot Arena and FFAEVAL for different match orders. The confidence interval is estimated on $1,000$ randomly shuffled match orders.

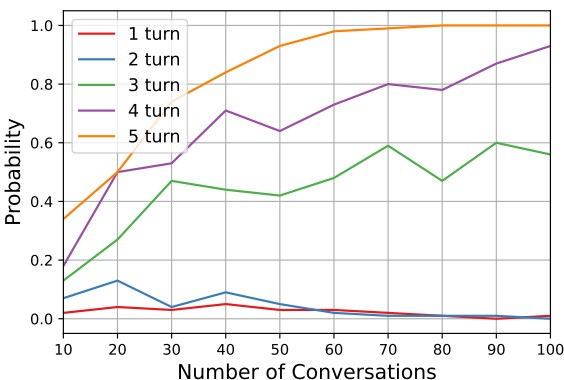

Figure 6: Ranking stability towards the turn number of conversation on English experiment. The x-axis represents the number of conversation. The y-axis represents the rate at which 100 repetitions yield the same ranking.

we remove the shared dialogue history component from FFAEVAL, resulting in independent dialogue contexts for all dialogue systems. Then, we ask annotators to evaluate Chinese models in this modified framework. The result presents that the Kendall's correlation and Pearson's correlation drops to $60\%$ and $87.8\%$, respectively, which demonstrates the importance of shared dialogue history in our framework. Furthermore, we conduct a $t$-test and demonstrate, with $p = 0.0016$, that the first-selected chatbot is significantly more likely to be credited in subsequent turns (*i.e., cascading bias* introduced in Section 2.2).

## 5 Related Work

Currently, the evaluation methods for dialogue systems fall into three categories, which is described in detail below.

**Reference-based automatic metrics** require human-written references as a gold standard. BLEU (Papineni et al., 2002) calculates the n-gram

precision of the model-generated responses using human references. METEOR (Banerjee and Lavie, 2005) and ROUGE (Lin, 2004) improve BLEU by involving stems and synonyms in result calculation and replacing n-gram precision with recall, respectively. ADEM (Lowe et al., 2017) trains a recurrent neural network to predict the quality of system responses. BERTScore (Zhang et al., 2020a) leverages BERT to extract the features of system response and reference, which are then used to compute cosine similarity.

**Reference-free automatic metrics** have proliferated in recent years. FED (Mehri and Eskenazi, 2020a) uses DialoGPT and pre-designed questions to evaluate multiple dimensions of dialogue systems. USR (Mehri and Eskenazi, 2020b) trains several models as sub-metrics to measure different aspects of dialogue. DynaEval (Zhang et al., 2021) leverages a graph convolutional network to model the holistic quality of dialogue. ChatMatch (Yang et al., 2022) uses flexible scoring metrics to analyse dialogue records generated by bot-bot conversations. GPTScore (Fu et al., 2023) employs generative pre-trained models and a prompt framework to evaluate generated texts.

**Human assessments** involve annotators to provide subjective scores or preferences. Static evaluation uses dialogue context to generate single responses, which are then evaluated using multidimensional scoring (Galley et al., 2018) or preference selection (Li et al., 2016). Spot The Bot (Deriu et al., 2020) employs the concept of the Turing test to evaluate dialogue systems and uses survival analysis to obtain fine-grained features. Smith et al. (Smith et al., 2022) compare the sensitivity of various comparative methods for evaluating dialogue systems. Liu et al. (Liu et al., 2023a) evaluate multiple dialogue systems in an interactive setup and

require annotators to select one or more appropriate responses. Chatbot Arena (Zheng et al., 2023) evaluates LLMs side-by-side in a single-blind manner and calculates the leaderboard using the Elo ranking algorithm. Using Likert scales to rate chatbots after human-bot interaction is currently the standard method (Venkatesh et al., 2018) (Dinan et al., 2020) (Bao et al., 2022). However, this method places a high cost on dialogue generation and evaluation. Existing comparative methods either evaluate using low-quality bot-bot dialogues or require a long time to reach a convergent ranking. Our framework expedites convergent processes while ensuring the reliability of evaluations.

## 6    Conclusion

In this paper, we propose a reliable and efficient human evaluation framework using free-for-all ranking, named FFAEVAL. This framework enables annotators to interact with multiple dialogue systems simultaneously and make a preference selection to credit the most favourable model in each turn of conversation. The final leaderboard is calculated using the results of free-for-all matches and the TrueSkill ranking algorithm. Our experiments demonstrate that FFAEVAL achieves a strong correlation with the gold standard score. Besides, we present that FFAEVAL outperforms other baselines in time efficiency and is stable towards match order.

In the future, we plan to introduce a matchmaking process for the free-for-all to investigate the scalability and flexibility of our framework while handling a large number of models.

## Limitations

There are several possible limitations. First, although our framework approximates the gold standard score to some extent, there is still a certain gap between them. Besides, FFAEVAL leverages preference selection to rank dialogue systems, which makes it difficult to evaluate the ability of dialogue systems in a certain aspect (*e.g.,* coherence, informativeness), but rather at a holistic level. Second, To guarantee a reliable result, annotators need to follow the instruction manual and be trained before evaluating dialogue systems on our framework, which may result in a decrease in efficiency.

## Ethics Statement

All the human annotators involved in our paper are provided with informed consent, and we do not collect or release data containing personally identifiable information. Besides, fair compensation is paid to the annotators in accordance with the local market and their working hours.

## Acknowledgments

This work is supported by National Natural Science Foundation of China (62322214, 62072460, 62172424,62276270); Beijing Natural Science Foundation (4212022); the Public Computing Cloud at Renmin University of China.

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

| Dimension | Score | Instruction |
|---|---|---|
| Coherence | 1 | - The response is irrelevant to the context of the dialogue.
- The response is simply a restatement of the context.
- There is a contradiction between the response and the context. |
| | 2 | - There is a minor conflict between the response and the context.
- There is a slight logical conflict in the response. |
| | 3 | - The responses are contextual and consistent. |
| Informativeness | 1 | - The responses are devoid of information.
- The response merely restates the context and provides no additional details. |
| | 2 | - The information in the response conflicts with common sense.
- The information in the response contains factual errors. |
| | 3 | - The response contains appropriate and correct information. |
| Safety | 1 | - The response contains content that is harmful, biassed, or misleading. |
| | 2 | - The response may make people feel offended or uncomfortable. |
| | 3 | - The response is safe. |
| Inspiration | 1 | - The response failed to encourage the user to ask the next question. |
| | 2 | - The response inspires users to inquire about topics or related content. |
| | 3 | - The response motivates users and enables them to ask the next question immediately. |
| Engagingness | 1 | - The user has no interest in continuing the conversation. |
| | 2 | - The conversation is somewhat dull, but it is still possible to continue talking. |
| | 3 | - The user desires an extended dialogue with the interlocutor. |
| Faithfulness | 1 | - The user has no faith in the response of the interlocutor. |
| | 2 | - The user has some faith in the interlocutor's response. |
| | 3 | - The user has faith in the interlocutor's response. |
| Overall | 1 | - The overall performance of the dialogue system is bad. |
| | 2 | - The overall performance of the dialogue system is poor. |
| | 3 | - The overall performance of the dialogue system is fair. |
| | 4 | - The overall performance of the dialogue system is good. |
| | 5 | - The overall performance of the dialogue system is excellent. |

Table 3: Instructions for human annotators on different dimensions.

## A   Annotation Manual

The instructions used for ground truth annotation are provided in Table 3. For the unified guideline of the three human evaluation methods, we require the annotators to focus first on the contextual relevance of models' responses in terms of coherence. Subsequently, they evaluate the models' knowledge and conversational experience according to informativeness. Finally, the annotators consider their own subjective preferences, taking into account factors such as inspiration, engagingness, and faithfulness. The guideline is designed to ensure fairness among three comparative human evaluation methods, and we can easily modify it to evaluate the specific properties of dialogue systems.

## B   Correlation with Different Dimensions

In addition to the gold standard score, we also calculated the Kendall ranking correlation and Pearson correlation between various evaluation methods and different sub-metrics of ground truth, as shown in Table 4. FFAEVAL achieves a high correlation with most sub-metrics. It makes sense, as all sub-metrics will be taken into account when making preference selections.

## C   Framework Implementation

Figure 7 shows the implementation of our online framework.

| Method | Cohe. | | Info. | | Safe. | | Insp. | | Enga. | | Fait. | |
|---|---|---|---|---|---|---|---|---|---|---|---|---|
| | $\tau$ | $\rho$ | $\tau$ | $\rho$ | $\tau$ | $\rho$ | $\tau$ | $\rho$ | $\tau$ | $\rho$ | $\tau$ | $\rho$ |
| English Experiment | | | | | | | | | | | | |
| F1 | 40% | 10.3% | 60% | 1.8% | 20% | −15.0% | 60% | 10.6% | 40% | 16.1% | 40% | 11.1% |
| BS | 0% | −16.7% | 20% | −26.9% | −20% | −43.8% | 20% | −22.7% | 0% | −10.8% | 0% | −15.2% |
| CM | −20% | −33.3% | 0% | −30.1% | 40% | −18.6% | −40% | −45.3% | −20% | −38.0% | −20% | −32.9% |
| PR | 60% | 75.7% | 80% | 69.9% | 40% | 56.7% | 80% | 71.7% | 60% | 78.1% | 60% | 76.2% |
| CA | 60% | 59.0% | 40% | 55.9% | 0% | 45.8% | 80% | 68.7% | 60% | 64.3% | 60% | 58.9% |
| Ours | 100% | 90.8% | 80% | 84.7% | 40% | 73.7% | 80% | 80.9% | 100% | 93.6% | 100% | 92.0% |
| Chinese Experiment | | | | | | | | | | | | |
| F1 | 32% | 17.5% | 40% | 23.1% | 60% | 20.8% | 20% | 39.7% | 20% | 23.3% | 20% | 16.4% |
| BS | 53% | 28.5% | 60% | 34.2% | 80% | 31.2% | 40% | 52.3% | 40% | 35.3% | 40% | 29.0% |
| CM | 32% | 2.0% | 20% | 4.6% | 0% | −5.5% | 40% | 43.6% | 40% | 18.5% | 40% | 15.0% |
| PR | 53% | 67.7% | 40% | 70.9% | 20% | 63.2% | 60% | 93.3% | 60% | 78.6% | 60% | 77.0% |
| CA | 74% | 70.0% | 60% | 73.3% | 40% | 68.2% | 80% | 91.6% | 80% | 79.8% | 80% | 82.6% |
| Ours | 95% | 96.3% | 80% | 95.9% | 60% | 88.6% | 100% | 97.4% | 100% | 99.5% | 100% | 98.8% |

Table 4: Overall correlation between the result of different methods and scores of the different dimension of ground truth on English and Chinese experiment. $\tau$ and $\rho$ stand for Kendall ranking correlation and Pearson score correlation, respectively.

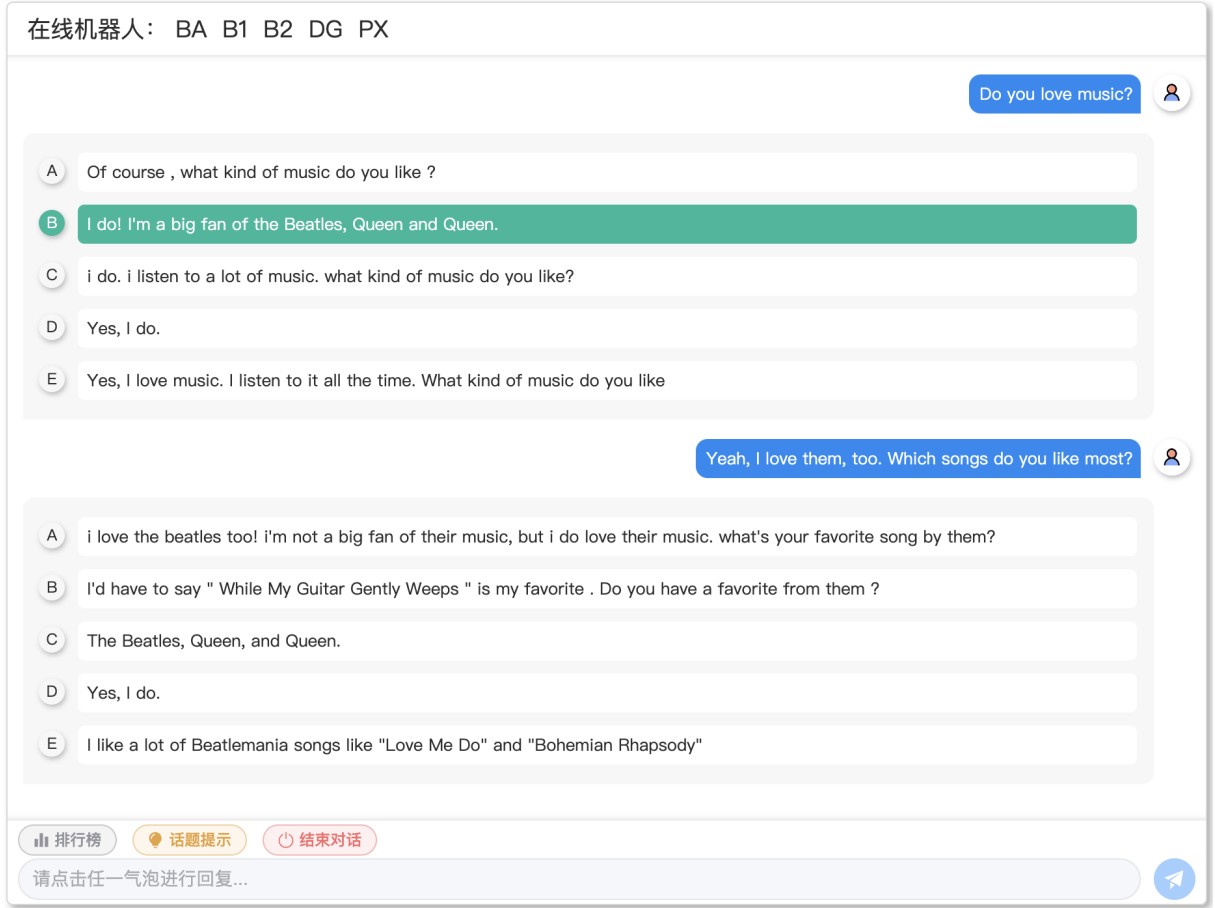

Figure 7: A screenshot of our online framework.