# OpenReview forum: "FFAEval: Evaluating Dialogue System via Free-For-All Ranking"
_EMNLP/2023/Conference — EMNLP 2023 Findings_

### Official Review · Reviewer_giRW · 2023-08-04

**Soundness:** 3

**Excitement:**

2: Mediocre: This paper makes marginal contributions (vs non-contemporaneous work), so I would rather not see it in the conference.

**Paper Topic And Main Contributions:**

This paper addresses the issue of comparing the performance of different open-domain dialogue systems. Automatic evaluation metrics have poor correlation with human evaluations in dialogue generation tasks. Human evaluations, involving multi-dimensional scores from annotators, are credible but time-consuming. Therefore, the authors propose a human evaluation framework called FFAEVAL, which adopts a Free-For-All Ranking method, aiming to provide a reliable and efficient evaluation method.

By sharing dialogue history, this framework allows evaluators to conduct single-blind, multi-turn dialogues with multiple dialogue systems at the same time. In the ensuing free-for-all competition, evaluators can choose the model they think is the best from all participating dialogue systems in each round. Finally, the performance of each model is represented by calculating TrueSkill scores.

**Questions For The Authors:**

A. Have you ever tried some current dialogue systems and compare the ranking order with ChatBot Arena.

B. What is the background of the human evaluators. Are they proficient in Chinese and English?

C. Is there a data breach risk using the topics in the Daily Dialog dataset? Since some models may are trained on it but others may not.

D. What is the backbone model of the BERTScore metric? And why use the F1 score instead of the BLEU score?

**Reasons To Accept:**

1. The authors propose to compare different dialogue system via a Free-For-All match.
2. The proposed framework: FFAEVAL can compare multiple dialogue system at meantime, which can improve the efficiency of the evaluation system.
3. Dialogue evaluation is an important topic in LLMs era. Besides, with the popularity of open domain dialog systems like ChatGPT and their need for human evaluation, it becomes more and more important to design an effective and efficient evaluation system.

**Reasons To Reject:**

1. The proposed method may be less relevant to the authors' motivations in abstract section (automatic scores are not effective and human evaluation scores are not affordable). Since the proposed framework FFAEVAL and some similar framework like Chatbot Arena are used to do comparison between dialogue systems, I do not think it can be directly used to evaluate a single dialogue system, like give a fluency score or something like that. So these arena-based evaluation systems may not solve the problems of current score-based evaluation systems.
2. The claims of these paper lack solid experiments support. Dialogue systems evaluated in experiments are kind of out-dated, such as DialoGPT and PLATO-XL. The poor performance of old models may influence the evaluation process. Using current LLM-based chatbot like alpaca, vicuna may be more convincing. Besides, the experiments section lacks some details, like the number of test examples and the inference settings.
3. Although the authors state some difference between their framework and Chatbot Arena (a popular evaluation framework for comparing different dialogue systems), I think the difference between the two methods is not obvious. Chatbot Arena compares models two by two and then computes the elo score. FFAEVAL compares all models in one conversation and then computes the TrueSkill score.

**Reproducibility:**

3: Could reproduce the results with some difficulty. The settings of parameters are underspecified or subjectively determined; the training/evaluation data are not widely available.

**Reviewer Confidence:**

4: Quite sure. I tried to check the important points carefully. It's unlikely, though conceivable, that I missed something that should affect my ratings.

---

> ### Author Rebuttal · Authors · 2023-08-29
>
> We appreciate your reviews and address your concerns as follows.
>
> > Comment 1 in Reason to Reject: The proposed method may be less relevant to the authors' motivations in abstract section (automatic scores are not effective and human evaluation scores are not affordable). Since the proposed framework FFAEVAL and some similar framework like Chatbot Arena are used to do comparison between dialogue systems, I do not think it can be directly used to evaluate a single dialogue system, like give a fluency score or something like that. So these arena-based evaluation systems may not solve the problems of current score-based evaluation systems.
>
> Thank you for your comment. FFAEval is indeed incapable of evaluating and scoring a single model. However, in most cases, the goal of the evaluation methods is to identify differences in capabilities between models to better guide researchers or users in discovering or selecting models that perform the best. Despite the fact that score-based metrics can be used to evaluate and score a single model, they are rarely employed for this purpose. As the goal of both score-based methods and arena-based methods is the same, we believe FFAEval can be an alternative to score-based methods to some extent.
>
> > Comment 2 in Reason to Reject: The claims of these paper lack solid experiments support. Dialogue systems evaluated in experiments are kind of out-dated, such as DialoGPT and PLATO-XL. The poor performance of old models may influence the evaluation process. Using current LLM-based chatbot like alpaca, vicuna may be more convincing. Besides, the experiments section lacks some details, like the number of test examples and the inference settings.
>
> Thank you for your comment. Our goal is to verify the reliablity of FFAEval and we need a ground truth ranking as a reference. However, it's difficult for us to obtain a ground truth ranking for the current LLM as an acknowledged evaluation method is still under-investigated. Therefore, we use open-domain dialogue systems (e.g., PLATO-XL), as we can find an acknowledged and frequently used method (i.e., score-based human evaluation including overall, coherence, informativeness, etc.) by researchers (e.g., [1][2][3]).
>
> Thanks for pointing out this. We will add more detailed information about the experimental setup in our next revision. We use 500 turns of conversation as test examples. To ensure fairness, we set the temperature to 1 for each dialogue system, and the length of input and output depends on the maximum length supported by the model.
>
> > Comment 3 in Reason to Reject: Although the authors state some difference between their framework and Chatbot Arena (a popular evaluation framework for comparing different dialogue systems), I think the difference between the two methods is not obvious. Chatbot Arena compares models two by two and then computes the elo score. FFAEVAL compares all models in one conversation and then computes the TrueSkill score.
>
> Thank you for your comment. The difference between FFAEval and Chatbot Arena is as follows:
>
> * FFAEval maintains a shared dialogue history, while Chatbot Arena does not. Referring to chapter 4.5 in the paper, shared dialogue history is the key for FFAEval to achieve better performance by a large margin. This shared dialogue history serves as a critical component, ensuring a fair comparison between models during multi-turn conversations.
> * The annotation of FFAEval is more fine-grained. In Chatbot Arena, the annotator annotates one time after the conversation ends, while in FFAEval, the annotator annotates each turn and calculate the result based on this per-turn annotation. The evaluation process of FFAEval is more robust and reliable as the influence of incorrect annotation is less.
> * The ranking algorithm is different. FFAEval uses the TrueSkill algorithm to calculate the result of a free-for-all match, while Chatbot Arena uses the Elo algorithm to calculate the result of a 1v1 match. Thanks to the TrueSkill algorithm and free-for-all match, FFAEval converges faster and achieves better time efficiency. Besides, using TrueSkill mitigates the influence of model competition order, as demonstrated in Figure 5 of the paper.
>
> > Question A: Have you ever tried some current dialogue systems and compare the ranking order with ChatBot Arena.
>
> Thanks for your question. According to the answer to Comment 2, our goal is to verify the reliability of FFAEval and it's difficult for us to obtain a ground truth ranking for the current LLM as an acknowledged evaluation method is still under-investigated. Therefore, we have not tried the current LLMs.
>
> > Question B: What is the background of the human evaluators. Are they proficient in Chinese and English?
>
> Thanks for your question. Referring to L331, the annotators are all Chinese undergraduate students. Their native language is Chinese, and their English is at Chinese College Entrance Examination level. They are able to read and understand English conversations fluently.
>
> > Question C: Is there a data breach risk using the topics in the Daily Dialog dataset? Since some models may are trained on it but others may not.
>
> Thanks for your question. We think the impact of data breach risk is minimal for the following two reasons:
>
> * The sampled starting utterance is not directly used for training. And we rewrite the sampled sentences to help annotators and chatbots create more meaningful conversations.
> * Although the initial starting utterances are sourced from the Daily Dialog dataset, it's important to note that in the following conversations, annotators are unlikely to engage with chatbots in accordance with the data distribution found in Daily Dialog training set.
>
> > Question D: What is the backbone model of the BERTScore metric? And why use the F1 score instead of the BLEU score?
>
> Thanks for your question about experiment details. We use roberta-large and bert-base-chinese as the backbone of BERTScore for evaluating English and Chinese models, respectively. BLEU is a precision-focused metric, while F1 considers both values of precision and recall. Therefore, we use the F1 score as one of the representative methods of reference-based metrics.
>
> *Reference:*
>
> [1] Roller, Stephen, et al. "Recipes for Building an Open-Domain Chatbot." Proceedings of the 16th Conference of the European Chapter of the Association for Computational Linguistics: Main Volume. 2021.
>
> [2] Bao, Siqi, et al. "PLATO: Pre-trained Dialogue Generation Model with Discrete Latent Variable." Proceedings of the 58th Annual Meeting of the Association for Computational Linguistics. 2020.
>
> [3] Yu, Jifan, et al. "XDAI: A Tuning-free Framework for Exploiting Pre-trained Language Models in Knowledge Grounded Dialogue Generation." Proceedings of the 28th ACM SIGKDD Conference on Knowledge Discovery and Data Mining. 2022.

---

### Official Review · Reviewer_nRyd · 2023-08-04

**Typos Grammar Style And Presentation Improvements:** Line 428 - “analysis” -> “analyze”
**Soundness:** 4

**Excitement:**

2: Mediocre: This paper makes marginal contributions (vs non-contemporaneous work), so I would rather not see it in the conference.

**Missing References:**

[1] Liu, Sijia, et al. "Towards credible human evaluation of open-domain dialog systems using interactive setup." Proceedings of the AAAI Conference on Artificial Intelligence. Vol. 37. No. 11. 2023.

[2] Smith, Eric, et al. "Human Evaluation of Conversations is an Open Problem: comparing the sensitivity of various methods for evaluating dialogue agents." Proceedings of the 4th Workshop on NLP for Conversational AI. 2022.

**Paper Topic And Main Contributions:**

Topic: The paper studies the topic of human evaluation of dialogue systems.

Research problem: Given the poor reliability of automatic metrics, human evaluation remains the gold standard in assessing dialogue systems. Yet, existing human evaluation setups, such as multi-dimensional scoring based on instruction manuals, pairwise comparisons of dialogue systems, are time-consuming.

Contributions: The authors propose a Free-For-All ranking approach to interactively evaluate the dialogue systems. Specifically, a human user will simultaneously chat with multiple dialogue models and select the best response among all the candidates generated by the models. The conversation continues with the selected response.

**Questions For The Authors:**

1. How does FFAEVAL deal with the case when none of the generated responses are appropriate [1]?

2. Line 302, does comparison with reference-based metrics make sense? My understanding is that FFAEVAL is designed for the interactive evaluation setting. As the conversation evolves, the trajectory will diverge from the original human-human conversations.

3. In Figure 1 and Table 1, considering that you are ranking just five dialogue models, how reliable or statistically significant is your findings? Additionally, if the number of models increases, will the reliability of the human user decrease given that they have to choose among a large pool of response candidates?

**Reasons To Accept:**

1. The clarity of the paper is good.

2. The proposed method is technical sound and the analyses are comprehensive

**Reasons To Reject:**

The design of FFAEVAL is not new. Refer to [1] in the missing reference, which contains two similar interactive evaluation setups as FFAEVAL: (1) multi-model Select-One-Best-from-All and (2) multi-model Select-All That-Apply.

**Reproducibility:**

4: Could mostly reproduce the results, but there may be some variation because of sample variance or minor variations in their interpretation of the protocol or method.

**Reviewer Confidence:**

4: Quite sure. I tried to check the important points carefully. It's unlikely, though conceivable, that I missed something that should affect my ratings.

---

> ### Author Rebuttal · Authors · 2023-08-29
>
> We appreciate your reviews and address your concerns as follows.
>
> > Comment 1:  The design of FFAEVAL is not new. Refer to [1] in the missing reference, which contains two similar interactive evaluation setups as FFAEVAL: (1) multi-model Select-One-Best-from-All and (2) multi-model Select-All That-Apply.
>
> Thank you for pointing out the missing reference. However, we notice that there are  differences between FFAEval and [1], as follows:
>
> * The ranking algorithm differs between [1] and FFAEval. Whereas [1] calculates the winning rate, FFAEval models the evaluation process as free-for-all matches using the TrueSkill ranking algorithm.  We change the ranking algorithm to winning rate in our evaluation method and find that the required chatbot response for evaluation convergence increases from 875 to 1250, illustrating the faster convergence by our ranking method.
> * Since we choose the TrueSkill, we initially generate the ranking result of a multi-turn conversation. Subsequently, we use this ranking result to update the overall leaderboard by TrueSkill. In contrast, [1] directly employs the result of a single turn to update the overall leaderboard.
>
> Besides the difference in framework design, it's crucial to highlight that [1] did not compare with any other evaluation baselines. Conversely, our approach comprehensively compares the proposed FFAEval with laterative evaluation baselines, which validates the reliability of this interactive evaluation framework.
>
> > Question 1: How does FFAEVAL deal with the case when none of the generated responses are appropriate [1]?
>
> That's a good question and we have also considered this circumstance. To solve this problem, our online system enables annotators to regenerate dialogue system responses if none of them are appropriate.
>
> > Question 2: Does comparison with reference-based metrics make sense?
>
> Thanks for your question. As our goal is to demonstrate the reliability of FFAEval in a fair and comprehensive manner, we believe it is essential to use a variety of evaluation methods as baselines. Since reference-based metrics are still frequently used as a reference for open-domain dialogue tasks, we think it makes sense to use them as baselines. Except for the reference-based method, we also compare FFAEval with reference-free metrics and various human evaluation methods, demonstrating the comprehensiveness of our experiment.
>
> > Question 3: How reliable or statistically significant is the result in Table 1? If the number of models increases, how to handle a large pool of response candidates?
>
> Thanks for your question. The results in Table 1 are statistically significant, please refer to official comment for more details.
>
> If we need to evaluate a large number of models, we can modify the framework to randomly select a fixed number of dialogue systems to be evaluated before a conversation begins. This modification can solve the difficulty of selecting the best response among a large number of candidates.

---

### Official Review · Reviewer_ojyE · 2023-08-11

**Typos Grammar Style And Presentation Improvements:** A. L209 add a citation for cascading …
**Soundness:** 3

**Excitement:**

3: Ambivalent: It has merits (e.g., it reports state-of-the-art results, the idea is nice), but there are key weaknesses (e.g., it describes incremental work), and it can significantly benefit from another round of revision. However, I won't object to accepting it if my co-reviewers champion it.

**Paper Topic And Main Contributions:**

The authors propose a reliable and efficient human evaluation framework "FFAEval". The framework is used to evaluate and compare multiple open-domain dialogue systems through the usage of free for all ranking approach that allows annotators to pick the most favourable model in each turn and continue the conversation.

The proposed framework has been used to evaluate the existing model that have been trained on English and Chinese datasets. Results obtained using this framework shows higher level of correlation with human assessment compared to other evaluation methods.

**Questions For The Authors:**

A. In order to get the GS score, Does all the models get a random conversation to begin? If all the models get a random conversation, how will that ensure fairness in comparison?
B. What are the sub-metrics used? L357 just mentions various sub-metrics
C. Please add more information as to why reference-free metrics do not succeed on this task.
D. Are the results in Table 1 statistically significant?
E. Did multiple annotators annotate the same conversation? What is the IAA score? How many groups of annotators were asked to annotate the GS score.
F. How was the time for annotation calculcated? Were all the experiements to annotate done with the same samples?

**Reasons To Accept:**

A. Tackles an important problem of improving evaluation process for dialogue systems through an interesting framework
B. Experiments conducted on multiple languages to demonstrate the effectiveness of the proposed evaluation framework.

**Reasons To Reject:**

A. Lack of information regarding the annotation guidelines provided for annotation deployed dialogue systems. What are the criteria the user is looking for during annotation?
B. The proposed system/framework might only capture which dialogue system is better for a users preference but does not provide insight into specific properties of the dialogue system, like whether has issues with coherency, or hallucinates. The framework feels incomplete without being able to understand the nuances of the each model.


**Reproducibility:**

2: Would be hard pressed to reproduce the results. The contribution depends on data that are simply not available outside the author's institution or consortium; not enough details are provided.

**Reviewer Confidence:**

4: Quite sure. I tried to check the important points carefully. It's unlikely, though conceivable, that I missed something that should affect my ratings.

---

> ### Author Rebuttal · Authors · 2023-08-29
>
> Thank you for your valuable reviews! Your concerns about experiment details are meaningful for our work. We would like to address your questions in the following part and include these details in our next revision.
>
> > Comment A in Reason to Reject: Lack of information regarding the annotation guidelines provided for annotation deployed dialogue systems. What are the criteria the user is looking for during annotation?
>
> Thank you for pointing out this. Due to page limitations, we supplement our annotation manual in Appendix A. To ensure consistency among the three human evaluation methods and the overall metric of GS score, the annotation process follows the same unified guidline as provided.
>
> In particular, to better control variables during selecting the preferable response, we require the annotators to focus first on the contextual relevance of models' responses in terms of  coherence. Subsequently, they evaluate the models' knowledge and conversational experience according to informativeness. Finally, the annotrators consider their own subjective preferences, taking into account factors such as inspiration, engagingness, and faithfulness.
>
> To avoid any potential misunderstandings, we would like to highlight this information in the main paper after revision and supplement more description on annotation details.
>
> > Comment B in Reason to Reject: The proposed system/framework might only capture which dialogue system is better for a users preference but does not provide insight into specific properties of the dialogue system, like whether has issues with coherency, or hallucinates. The framework feels incomplete without being able to understand the nuances of the each model.
>
> Thanks for your comment! Similar to our response to your previous comment (Comment 1), annotators are required to use our framework following a unified guideline. So we capture not only the subjective user preference but also the overall performance of the dialogue system, including coherence, informativeness, etc.
>
> For the evaluation of specific properties, as the framework is a type of comparative system, we only need to modify the annotation guidelines to match the definition of the property we want to evaluate.
>
> > Question A: In order to get the GS score, does all the models get a random conversation to begin? If all the models get a random conversation, how will that ensure fairness in comparison?
>
> Thank you for your question. Indeed, the starting utterance for the conversation is an important control variable, which is taken into consideration in our experiments.
>
> In particular, on each occasion, we begin by randomly selecting an initial utterance. This utterance is subsequently employed by all dialogue systems to initiate human-bot conversation. Therefore, the comparison is relatively fair, as the starting utterances of each model is the same.
>
> > Question B: What are the sub-metrics used? L357 just mentions various sub-metrics.
>
> We apologize for any confusion that the term ``sub-metrics`` may cause. The ``sub-metrics`` are consistent with the ``other dimensions`` mentioned in L358, which refer to the dimensions except overall, including coherence, informativeness, safety, inspiration, engagingness, and faithfulness. A detailed introduction of these dimensions can be found in Appendix A.
>
> > Question C: Why reference-free metrics do not succeed on this task?
>
> That's a good question and we will provide more supplements in our next revision. The reason for the poor performance, in our opinion, is twofold:
>
> * *Reference-free metrics, such as ChatMatch, may rely on evaluating bot-bot converstaions, while the quality degradation of dialogue between bots happens easily.* Although we generate 100 rounds of dialogue following the setting in the original paper, the content still lacks meaningful information on various topics, and duplicate responses frequently occur. This may prevent some powerful models from demonstrating their abilities in conversations. We provided an example of a conversation between DialoGPT and PLATO-XL:
>
> ```
> DialoGPT: No problem, good luck with your future endeavors!
> PLATO-XL: Thanks, you too !
> DialoGPT: You're welcome :D
> PLATO-XL: Have a great day!
> DialoGPT: Thank you so much! You have an awesome day as well!
> PLATO-XL: You're welcome!
> ```
>
> * ChatMatch employs multiple automatic metrics, such as perplexity and distinct-n, as well as rule-based algorithms such as the diversity scoring algorithm, to assess low-quality bot-bot dialogues. Unlike established reference-based metrics such as  F1 and BERTScore, which assess the bot's outputs by contrasting them with the high-quality human-written references, it is understandable that ChatMatch might yield relatively low performance.
>
> > Question D: Are the results in Table 1 statistically significant?
>
> Thanks for your question. The results in Table 1 are statistically significant, please refer to official comment for more details.
>
> > Question E: Did multiple annotators annotate the same conversation? What is the IAA score? How many groups of annotators were asked to annotate the GS score.
>
> Thanks for your questions about experiment details. As mentioned in L360, each conversation is assigned to three annotators for annotation. So three annotators annotate the same conversation.
>
> We use Pearson's correlation to calculate inter-annotator agreement (IAA) score. The IAA scores for the English and Chinese experiments are 0.59 and 0.63, respectively. We believe the human evaluation is reliable, as the IAA score is around 0.6.
>
> Three annotators are organized into a single group. We evaluate the English model by four groups of annotators, and evaluate the Chinese model by another four groups. So we used a total of eight groups of annotators.
>
> > Question F: How was the time for annotation calculcated? Were all the experiements to annotate done with the same samples?
>
> Thanks for your question. For the English experiment, we use the same set of annotators to independently assess different human evaluation methods. For each human evaluation method, such as FFAEval, we record the entry time of each annotator into the online system. Annotators are then required to focus exclusively on the annotation task until completion. Then, we record the exit times of annotators from the system. Therefore, we can calculate the time spent by each annotator and obtain the total annotation time through summation. The same annotation process is used for the Chinese experiment.
>
> In particular, when designing the time efficiency experiment, we consider whether to record the time to annotate an equivalent number of conversations or to record the time requried to reach a stable score. We find that FFAEval attains a stable score with the fewest number of conversations, prompting us to employ this specific number of samples as the stopping criteria.

---

### Official Review · Reviewer_H8nE · 2023-08-12

**Soundness:** 4

**Excitement:**

4: Strong: This paper deepens the understanding of some phenomenon or lowers the barriers to an existing research direction.

**Missing References:**


In the Appendix, there's a need for citations to provide the sources of inspiration for defining the dimensions within each category for evaluating dialogue.

**Paper Topic And Main Contributions:**

The author introduced an automated evaluation metric that calculates Trueskill scores from diverse models. These scores are then employed to evaluate responses from a newly suggested model, selecting the optimal one. The study showcased a robust correlation between their metric and human-based scoring. This achievement was credited to the metric's prowess in managing history-based conversations, augmenting context-based learning reliability. Consequently, this enhancement led to a significant improvement in alignment with human evaluations. The paper additionally included a time comparison, substantiating time and convergence efficiency in comparison to human evaluation. A sensitivity analysis across various turns was conducted, contributing to a comprehensive assessment. Statistical representation further enhanced the clarity and impact of these analyses.

**Reasons To Accept:**

Reasons:
1. The study's outcomes demonstrated a strong match between their new metric and human evaluations. The metric's success in handling historical context was the key to this alignment.

2. The authors effectively used statistical graphics to compare their model's confidence levels with Chatbot Arena, enhancing their analysis.

3. Their investigation into the impact of multi-turn conversations on response quality provided valuable insights into the role of dialogue length.

4. While they clearly defined evaluation dimensions for each conversation, the study missed mentioning the sources that inspired these dimensions.
5. They mentioned ethical considerations for human annotation and privacy is maintained while collecting information.

**Reasons To Reject:**

The write-up is skillfully crafted, and I couldn't identify any shortcomings that would warrant rejection.

**Reproducibility:**

4: Could mostly reproduce the results, but there may be some variation because of sample variance or minor variations in their interpretation of the protocol or method.

**Reviewer Confidence:**

4: Quite sure. I tried to check the important points carefully. It's unlikely, though conceivable, that I missed something that should affect my ratings.

---

> ### Author Rebuttal · Authors · 2023-08-29
>
> We much appreciate your reviews and recognition of our work! Your concerns are addressed as follows:
>
> > While they clearly defined evaluation dimensions for each conversation, the study missed mentioning the sources that inspired these dimensions.
>
> Thanks for your questions about the details of evaluation dimensions. We follow the evaluation process of some dialogue systems to design these dimensions. In particular, we evaluate the overall score following [1], coherence, informativeness, and engagingness following [2], and safety following [3]. In addition, we introduce two new dimensions, inspiration and faithfulness, because we believe that the existing dimensions do not adequately capture all aspects of the model's capabilities. Inspiration is a turn-level metric measuring how inspiring the response of chatbot is and faithfulness is a dialog-level metric measuring whether the annotator has faith in the chatbot.
>
> *Reference:*
>
> [1] Bao, Siqi, et al. "PLATO: Pre-trained Dialogue Generation Model with Discrete Latent Variable." Proceedings of the 58th Annual Meeting of the Association for Computational Linguistics. 2020.
>
> [2] Bao, Siqi, et al. "PLATO-XL: Exploring the Large-scale Pre-training of Dialogue Generation." Findings of the Association for Computational Linguistics: AACL-IJCNLP 2022. 2022.
>
> [3] Bao, Siqi, et al. "PLATO-K: Internal and External Knowledge Enhanced Dialogue Generation." arXiv preprint arXiv:2211.00910 (2022).

---

### Meta-Review · Area_Chair_wi2Z · 2023-10-06

**Recommendation:** 5

**Metareview:**

This paper introduces a novel human evaluation framework, FFAEVAL, to compare the performance of various open-domain dialogue systems. Recognizing the limitations of automatic evaluation metrics and the time-consuming nature of traditional human evaluations, the authors propose a Free-For-All Ranking method that allows evaluators to conduct single-blind, multi-turn dialogues with multiple dialogue systems simultaneously. The paper offers a valuable contribution to designing effective and efficient evaluation methods for dialogue systems.

---

### Decision · Program_Chairs · 2023-10-07

**Decision:**

Accept-Findings

**Comment:**

This paper introduces a novel human evaluation framework, FFAEVAL, to compare the performance of various open-domain dialogue systems. Recognizing the limitations of automatic evaluation metrics and the time-consuming nature of traditional human evaluations, the authors propose a Free-For-All Ranking method that allows evaluators to conduct single-blind, multi-turn dialogues with multiple dialogue systems simultaneously. The paper offers a valuable contribution to designing effective and efficient evaluation methods for dialogue systems.